# Clinical Approaches to Nestorone Subdermal Implant Therapy in Women’s Health

**DOI:** 10.3390/biomedicines11092586

**Published:** 2023-09-21

**Authors:** Guilherme Renke, Consuelo Callizo, Raphaela Paes, Mariana Antunes, Glaycon Michels, Luana Concha, Ordânio Almeida, Christiane Valente, Thomaz Baesso, Bruna Giovannoni

**Affiliations:** 1Nutrindo Ideais Performance and Nutrition Research Center, Rio de Janeiro 22411-040, Brazildrglaycon@hotmail.com (G.M.);; 2Clementino Fraga Filho University Hospital, Federal University of Rio de Janeiro, Rio de Janeiro 21941-913, Brazil; 3Centro de Pesquisa e Assistência em Reprodução Humana, Salvador 40210-341, Brazil

**Keywords:** contraception, menopause, implant, Nestorone, segesterone acetate, hormone replacement therapy, hormone, neuroprotection

## Abstract

Segesterone acetate (SA) or Nestorone, a fourth-generation progestogen, is a synthetic compound with high progestational activity and no androgenic, glucocorticoid, or anabolic effects. However, due to its oral inactivity, SA must be used by other routes, such as subcutaneous. Thus, considering its peculiar properties, the SA subdermal implant is successfully used in female contraception and postmenopausal hormone replacement therapy (HRT). In recent years, its potential uses in endometriosis, polycystic ovaries syndrome (PCOS), and a new therapeutic possibility for neuroprotection have made this treatment extremely interesting. However, the absence of a standardized dose and the long-term safety of SA implant therapy in women is still controversial. Here, we present the possible indications, doses, limitations, and side effects of SA implant therapy.

## 1. Introduction

Segesterone acetate (SA), marketed under Nestorone, Elcometrine, ST1435, and Annovera, is a 19-nor steroid developed by the Population Council and introduced for medical use in 2000 [1,2]. Synthetic progestin blocks both the estrogenic and androgenic peaks, thus reducing elevated levels of these hormones in the ovulatory phase. These effects make its use attractive in patients with estrogen-dependent or androgen-dependent pathologies, and it can cause regression of diseases such as endometriosis, adenomyosis, and uterine myomatosis, without causing climacteric effects (Figure 1). It has highly selective binding to the progesterone receptor (PR) and is most commonly used as a contraceptive method [1,2,3]. However, it is not effective orally (bioavailability of only 10%) and must be administered parenterally via topical hydroalcoholic gels, vaginal rings, or subdermal implants. When administered parenterally, SA has 100 times greater progestational activity than progesterone (P) [3]. This increased activity is due to SA’s strong coupling and binding interaction with PR (same position as P), the additional stabilizing contacts that SA forms between the 17α-acetoxy and 16-methylene groups, and the PR ligand’s binding domain [3].

SA acts primarily as a high-affinity agonist for the PR and, like P, it does not bind to sexual hormone-binding globulin (SHBG) but to albumin. It does not bind significantly to the androgen receptor (AR), estrogen receptor, or mineralocorticoid receptor. Additionally, it has no estrogenic, androgenic, antiandrogenic, or anti-mineralocorticoid activity. Despite a significant affinity for the glucocorticoid receptor, it shows no glucocorticoid effect, or if any, only at exceptionally high doses. On the other hand, the binding affinity of SA for the AR is 500–600 times lower than testosterone (T) [3].

The SA dosage for parenteral ovulation inhibition has been reported at 150 mcg per day, while the endometrial transformation dosage has been reported at 600 mcg per cycle. Due to its potent and highly selective binding to the PR, SA may also have applications in postmenopausal HRT to offset estrogen-induced endometrial proliferation [1,4,5,6,7].

### 1.1. Subcutaneous SA Implant Therapy

Some ways to prescribe SA include transdermal gel, vaginal rings, and the application of silastic implants [1,2,3,4,5,6,7]. Implants were first described as an alternative treatment for HRT and menopausal symptoms in the 1950s [8]. Thus, the research on the long-term contraceptive efficacy of subdermal silastic implants containing P dates back to the early 1970s, led by the Brazilian researcher Elsimar Coutinho [9]. Since then, several studies have demonstrated that this route of administration provides a safety profile and effective therapy without the fluctuations in blood levels commonly seen with transdermal administration [10,11,12,13,14]. The administration of hormone implants has been used worldwide for decades and it has increased after the etonogestrel implant (Nexplanon) was disseminated worldwide as an effective and safe contraceptive method [15]. In some countries, such as Brazil, non-absorbable implants (Silastic) are widely used to treat menopause, contraception, and endometrioses (Figure 2) [16,17,18,19]. Another peculiarity of this implant is the absence of peaks in serum hormone levels, reducing the possibility of side effects and secondary reactions. Concerning the safety of subcutaneous implant therapy, an extensive 7-year retrospective study of 400,000 women and over 1,200,000 implant procedures shows a safety profile. The overall complication rate was <1%, and the continuation of treatment after two insertions was 93% (C.I. 90–95) [20].

Studies have shown that a releasing implant that confers serum levels of SA in the range of 20–50 pg/mL (56–139 pmol/L) effectively blocks ovulation. At low plasma concentrations, SA acts on the hypothalamus and pituitary gland to suppress gonadotrophin release; at higher doses, SA also has a direct effect on ovarian estradiol (E2) production, which further blocks the production of high levels of E2 needed to trigger the pre-ovulatory increase in luteinizing hormone (LH) [21,22].

#### Contraception

The SA implant consists of a single stick containing 50 mg of the substance, made in an outsourced pharmacy (ELMECO, Salvador, Brazil), in tubes of dimethylpolysiloxane (silicone) that controls the product’s release rate, capable of inhibiting ovulation for up to 6 months [23]. The produced SA implant is 4 mm long and 2.4 mm in diameter, molded into rods with a cylindrical core. Each core is covered by a release rate limiting membrane in silicone rubber tubes. The 75 mcg/day release rate proved safe and was confirmed in laboratory tests in preliminary studies [12,24,25].

Sivin et al. (2004) performed a 2-year trial of a single SA implant conducted at three Latin American centers, enrolling 300 women [25]. The safety, effectiveness, and acceptability of SA contraceptive implants were evaluated. Three pregnancies occurred, the last at 18 months of use. The trial was halted because no pregnancies were expected in the first 18 months. At that time, 224 women had completed at least 18 months of use, and 99 women had used the implant for more than 24 months. From curve fitting, the estimated corresponding day-specific release rate of a 92 mg SA implant at the end of 18 months was 24.9 mcg/day, suggesting that two implants of 50 mg could safely prevent pregnancy for up to 540 days [25].

Sterile Silastic implants are inserted through a 2 mm incision into the subcutaneous tissue of the upper gluteal area or lower abdomen through a small anesthetized incision using a sterile stainless-steel trocar or disposable trocar kit in a simple 5 min procedure [16,17,18,19,25]. After the end of treatment, implants may remain in the patient. However, removing Silastic implants is recommended due to their residual contraceptive action for up to 24 months [25]. A specialized medical team should do the procedure, especially if the treatment is continued.

Some adverse effects may occur with the use of the SA implant. During the first three months of use, 50% of patients bleed irregularly. Bleeding disorders accounted for up to half of all reasons given for discontinuing the use of the method. In some clinical studies, bleeding decreases to 30% in the second trimester. These side effects diminish with continued use, and many women experience regular bleeding patterns after six to nine months of use [24]. Despite bleeding disorders, most patients continue to use the method after 12 months (80.5%). Even in patients with dysmenorrhea, laboratory analyses of previous studies confirm that after 24 months, there is no fall in hemoglobin (<1%) or other significant side effects [25]. In hirsute patients, particularly those with PCOS, androgen blockade results in a substantial improvement of symptoms more rapidly than with the use of oral contraceptives. It can be used in breastfeeding women, and it has been found that the breastfed babies of women using it have no detectable trace of the progestin in their blood [21,26].

In addition to its use in hormonal implants, the FDA approved SA associated with estrogen in a vaginal contraceptive system in 2018. Marketed under the brand name Annovera, it is reusable for up to one year as a birth control method for women. An open-label pharmacokinetic study of SA and ethinyl estradiol vaginal contraceptive system in 39 women found rapid absorption of both steroids through the vaginal mucosa after insertion, with steady-state systemic levels of SA and ethinyl estradiol achieved by the fourth day of use [21]. The two hormones in the vaginal contraceptive system are delivered through a soft silicone-based ring containing a core of SA and ethinyl estradiol and a core of SA alone (103 mg of SA and 17.4 mg ethinyl estradiol). After an initial burst of steroid accumulation on the surface of the ring, the drug is steadily released to consistent blood levels (an average daily dose of 0.15 mg of SA and 0.013 mg of ethinylestradiol), in contrast to the daily fluctuations of steroids with oral contraceptives [21].

SA is also a great candidate as a male contraceptive, and in recent years, numerous studies have evaluated its benefit and applicability in men. The administration of SA via a transdermal gel has good tolerability, minimal side effects, and effectively inhibits ovulation. SA transdermal gel combined with T also decreases spermatogenesis in men without serious adverse effects, suggesting a potentially effective male contraceptive [12,27]. Using T-gel (100 mg) and SA-gel (8 mg) suppressed sperm concentration to <1 million/mL or azoospermia in 89% of men, compared with only 23% of men using T-gel. Serum gonadotropins (LH) and follicle-stimulating hormone (FSH) were suppressed rapidly. Gonadotropin hormone concentrations of >1 IU/L after four weeks of treatment were sufficient to prevent treatment failure (sperm concentration > 1 million/mL) with 97% sensitivity. Most failures were due to inconsistent or non-response use of the products rather than non-response to drug use [28,29,30]. The same results were found in another study using the same concentration of SA (8.3 mg) with a lower concentration of T (62.5 mg) in the gel [31]. When asked about the acceptability of the T + SA method, more than 50% of the participants reported being satisfied or extremely satisfied with the treatment [28,29]. The secondary transfer of T and SA is a limitation of the method. It occurs after intense skin contact with the gel application site. The secondary transfer is lessened by a shirt or bath barrier before contact [32]. The method was even controlled using fertility self-tests performed at home, similar to the results found in traditional methods performed in the laboratory [33]. However, despite the promising clinical research in this field, it still lacks safety and standard dosage to indicate SA gel as a male contraceptive. Subcutaneous implants for male contraceptives with SA or SA + T were not evaluated. Regarding this possibility, future studies should demonstrate the benefit of the implant concerning the transdermal gel since the implant provides greater stability and hormone release, the absence of risk of forgetting to use the medication, and the convenience of prolonged use.

### 1.2. Endometriosis

A review of 11 articles with 335 patients showed that etonogestrel and SA-releasing implants improved the visual analog scale (VAS) pain score for cyclic pelvic pain/dysmenorrhea (VAS at the beginning ranged from 6.1 to 7.5 cm and after treatment it was 1.7–4.9 cm, *n* = 121), pelvic pain is not cyclic (VAS was initially 7.2–7.6 cm and after treatment it was 2.0–3.7 cm, *n* = 96), and sexual intercourse is painful (VAS at baseline was 1.61–8.3 cm and it was 1.0–7.1 cm after treatment, *n* = 87) [1]. Symptomatic improvement with subcutaneous P implantation was comparable to treatment with reserve medroxyprogesterone acetate (DMPA; average baseline VAS 6.5 and after DMPA treatment 3.0 compared to 2.0 after treatment with the implant) or 52 mg of levonorgestrel releasing the intrauterine system (LNG-IUS; baseline cyclic and non-cyclic pain scores of 7.3 and 7.4, respectively, decreased to 1.9 and 1.9 after treatment with IUS-LNG). Improvements were also shown in the quality-of-life scores, vitality, physical pain, social and mental health, and sexual function. SA subdermal implants improved endometriosis-related pain symptoms and quality of life, providing an additional component in managing endometriosis [1].

For seven months, another trial evaluated the efficacy of three different SA doses (150, 200, and 400 mcg) of implants administered in 21 women with endometriosis to relieve pain, associated symptoms, and bleeding patterns [34]. After implant removal, the follow-up period was six months. Pelvic pain decreased significantly in response to the treatment in all groups but returned to pretreatment levels post-treatment. Bleeding and spotting were the most common side effects followed by hypoestrogenic, with no significant difference among the groups. No substantial changes in total, LDL, or HDL cholesterol serum levels were observed. Thus, all three doses of SA implant effectively alleviated endometriosis-related pain, providing an exciting option for treating endometriosis-related pain [34].

### 1.3. Menopause

Menopause is a natural and physiological process intrinsic to women’s lives. It is characterized by the decline of reproductive hormones since the ovaries stop producing E2 and P, thus marking the end of the woman’s reproductive phase. Clinically, it is expressed by the definitive cessation of menstruation and a sexual hormone deficit leading to several clinical and psychosocial manifestations, such as vasomotor disorders, atrophic vaginitis, osteoporosis, cardiovascular disease (CVD), urinary symptoms, and conditions related to sexual stimulation and desire, directly impacting women’s quality of life [35].

Menopausal women occupy much of the agenda of gynecologists who seek information, counseling, and treatment of problems at this stage of life. In this sense, many of them are indications for E2-based therapy. This therapy’s beneficial effects for treating menopausal symptoms are well-known. Micronized P or synthetic progestin is mandatory in postmenopausal hormone therapy for non-hysterectomized women. In such cases, a synthetic progestin, like SA, is necessary for endometrial tissue protection, thus balancing the proliferative effects of estrogen and reducing the risk of hyperplasia and endometrial cancer [35].

To alleviate menopausal symptoms, several approaches have been applied to HRT, the most prominent being the administration performed with progestogen implants under the skin in preventing and treating menopause and menopausal-induced lesions [36]. Over time, HRT has been controversial about risks related to CVDs, thromboembolic diseases, and some types of cancer (such as breast and cervical cancer). In addition, when it comes to hormone administration through implants, it enables a proposal for individualized management of the dose and type of hormone. However, in menopause, limited evidence about implants’ efficacy and risks exists. In addition, there is no consensus among compounding pharmacies, and there is less control of compounded hormones when compared to those available by the pharmaceutical industry. Tests to verify the validity of these implants are practically unfeasible, with possible errors not detected throughout the manufacturing process, exposing the method to criticism and misuse [19,20]. In addition, fixed hormonal doses available in the pharmacy may not suit the majority of patients, thus leading many physicians to opt for compounding pharmacies where there is the opportunity to customize the treatment, as is the case with subcutaneous implants.

Thus, in an attempt to minimize the effects of suppressing these hormones and, consequently, alleviate symptoms during menopause, several methods have been adopted for HRT, including subcutaneous hormonal implants [37]. These silastic devices contain hormone molecules inside and are characterized by the programmed release of these hormones, which begins shortly after their subcutaneous insertion until 6 to 12 months. Besides SA implant, several hormones are used commercially through this route, including T, E2, Gestrinone, and Nomegestrol Acetate [19,20,38].

SA implant exhibits comparable endometrial protective effect to micronized P. Because of its anti-androgen action, SA can relieve the symptoms of post-menopausal women with hyperandrogenism and is currently evaluated as a potential progestin in menopausal hormone therapy [3].

Thus, considering the multiple mechanisms of action and pharmacological qualities of SA, we could suggest it alone or in association with E2 in treating menopause (Figure 3). We recently published the importance of HRT with E2 in menopause, especially concerning CV protection [39]. It is well established that women have protection against hypertension and coronary heart disease during the reproductive phase of their lives, which disappears after menopause [40]. Therefore, it is inevitable that the female sex hormones, namely estrogen and P, are mainly responsible for the same. However, the exact mechanism of cardio-protection remains enigmatic [41]. It is also well-established that PR is found on blood vessels/arterial walls and the endothelial cells lining the walls [42]. The hypothesis that P plays a key role in CV health and disease was evaluated. A perspective of cardioprotection has been suggested as a function of P-induced respiratory alkalosis, decreased plasma ionic calcium concentration, and generalized vasodilation [41].

Thus, it is exceptional to think that SA, even at low doses in the form of an implant, could bring CV benefits due to its high-affinity characteristics with PR. Furthermore, hormone deficits during menopause are related to an increase in the occurrence of dementia and neurodegenerative diseases in these women. As discussed in recent studies, SA appears to have a neuroprotective effect that could favor postmenopausal women [43,44].

### 1.4. Advancing of SA as a Neuroprotective Drug

Neurological diseases are mostly debilitating and have few effective treatments described in the literature, making them the target of scientific research, especially in the last decade. Synthetic progestin, SA, has a neuroprotective potential because of its efficacy in binding to brain PRs [43]. PRs are found in the adult human brain performing different regulatory functions, which makes their activation optimistic in brain repair [44]. The SA molecule is a specific agonist that performs selective binding to PRs and non-significant binding to glucocorticoid receptors and androgen receptors. It expresses a potent connection to PRs, which provides the substance the characteristic of acquiring a high affinity even at low dosages, giving the drug a potential characteristic in neuroprotection. Its oral administration in animals or humans is inactive but potentially active when administered subcutaneously or transdermal [43,44].

Well-documented evidence shows that the central nervous system level P functions in mood modulation and inflammation control, acting on neural plasticity and cognition [24]. SA is a synthetic progestin that exhibits protective neurological effects in vitro and in vivo without adverse impact, which we did not observe in other synthetic progestins, such as MPA, which is one of the most prescribed progestogens by physicians. In contrast to SA, MPA binds to androgen and glucocorticoid receptors with 300 times greater affinity [24,43]. Some studies compare P with SA and show that it can promote remyelination of axons and promyelinating effects in mice, preventing neurodegeneration and stimulating neurogenesis [24].

Evidence suggests that acute exposure to synthetic progestins alone or combined with estrogen plays regulatory roles in the brain’s neurogenic and neuroprotective responses [45]. When chronically administered subcutaneously in mice, combining SA with estrogen can increase the expression of IGF-1 and IGF-1R levels (IGF-1 receptors), suggesting that IGF-1 is an essential regulator of SA and estrogen-induced neurogenesis [24].

An experiment with intranasal administration of SA at a dose of 0.08 mg/kg in mice with cerebral occlusion showed reduced ischemic injury in males and females. The intranasal route can be a possible strategy for reversing ischemic areas after a stroke [46]. There is consistent evidence of increased neural responses in vivo induced by progestins, and among them, SA stands out as the most promising. Thus, further research is needed to define the potential of SA in patients with neurodegenerative conditions and diseases [45].

### 1.5. Limitations and Adverse Effects

Although the amount of evidence is limited due to a lack of standardization of dosages, SA can be used in a wide range of conditions and by different routes (Table 1). The SA implant provides women with safe, effective, and acceptable contraception for up to 2 years. In previous studies, no serious adverse events were associated with the method. Menstrual disorders such as irregular bleeding and prolonged spotting are the main causes of treatment discontinuation, averaging 7 per 100 women/year. Disruptions attributed to non-menstrual medical conditions occurred at rates of less than 5 per 100 per year. As with other implants, headaches, weight gain, and reduced libido were frequently cited as events leading to discontinuation. The most common medical complaints unassociated with discontinuation were pelvic pain and vaginal discharge. Postmenopausal women may experience fewer side effects, specifically spotting, due to endometrial atrophy caused by hypoestrogenism.

Regarding body weight, an average increase of 1.7 kg per participant was previously observed (*p* < 0.05). However, body weight decreased for 25.3% of women, and 11.5% experienced no increase or decrease. In the blood pressure assessment, there was a decrease in mean systolic levels from 115.3 to 110.3 mmHg at the implant removal visit, while diastolic pressure decreased from 75.6 to 71.3 mmHg (*p* < 0.05) [25].

As previously demonstrated, no deleterious biochemical changes were observed in patients using the SA implant. Statistically significant changes after 1 and 2 years were that significant reductions (*p* < 0.05) in the abnormally high levels of total cholesterol, LDL, and calcium were observed in both years of use. Decreases in abnormally high values were observed for lactate dehydrogenase (year 2) and for total protein (year one only). The number of women with increased abnormal values for serum glutamic–oxaloacetic transaminase and serum glutamic–pyruvic transaminase was recorded in the first year, but abnormal values in these parameters when measured at the end of 2 years were not significantly different from the baseline [25].

As previously mentioned, the lack of a standardized drug containing SA, whether transdermal or subcutaneous, is a barrier to its use in both gynecology and general practice. Besides its potential therapeutic uses, this method with a half-year life should have special appeal to younger women, particularly those whose families are not complete and those with conditions such as endometrioses and PCOS.

## 2. Discussion

SA has unique characteristics that may be useful in several clinical conditions, such as endometriosis, contraception, post-menopause, and PCOS (Table 2). Recent studies show its possible additional neuroprotective effect, but no conclusive evidence remains. Despite using SA for many years, few countries use this substance in contraceptive methods or in treating women’s health. Our clinical experience demonstrates unique characteristics different from other progestins. This is especially because the SA implant confers similar clinical effects to combined oral pills, notably an important improvement in the skin, improvement in hair, and the absence of side effects such as edema or weight gain. Patients with severe PCOS or post-menopausal hyperandrogenism may benefit from this therapy with notably improved symptoms.

However, the ideal dose of the SA implant is still a question, especially in overweight and obese patients where the 50 mg twice-yearly dose has not yet been validated. Thus, the need for randomized clinical studies to validate its use in overweight patients, including comparing dosages (50 to 200 mg), may be extremely necessary to correctly personalize the use of the SA implant.

## 3. Conclusions

This review summarizes the safety, tolerability, and possible indications of the SA implant in a general population of women. Despite the lack of standardization for its use in some conditions, such as in menopausal women, continuation rates are excellent with minimal adverse effects. The CV and neuroprotective effects of SA appear to be favorable. However, the development of future studies of this hormone and its administration route seems necessary for the long-term evaluation of this HRT.

## Figures and Tables

**Figure 1 biomedicines-11-02586-f001:**
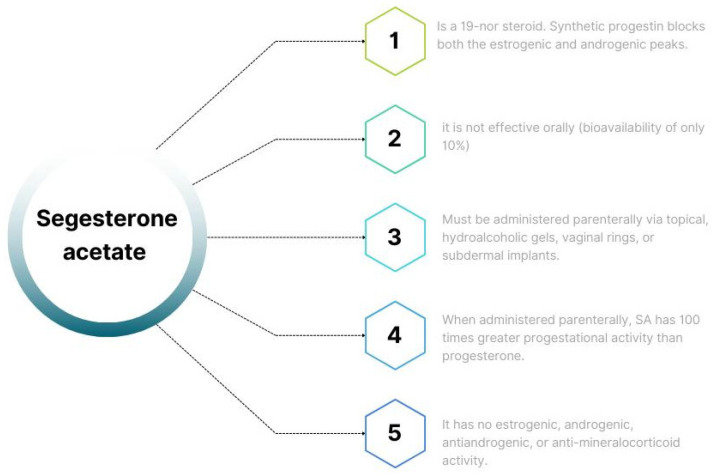
Characteristics of synthetic progestin SA. (SA: segesterone acetate).

**Figure 2 biomedicines-11-02586-f002:**
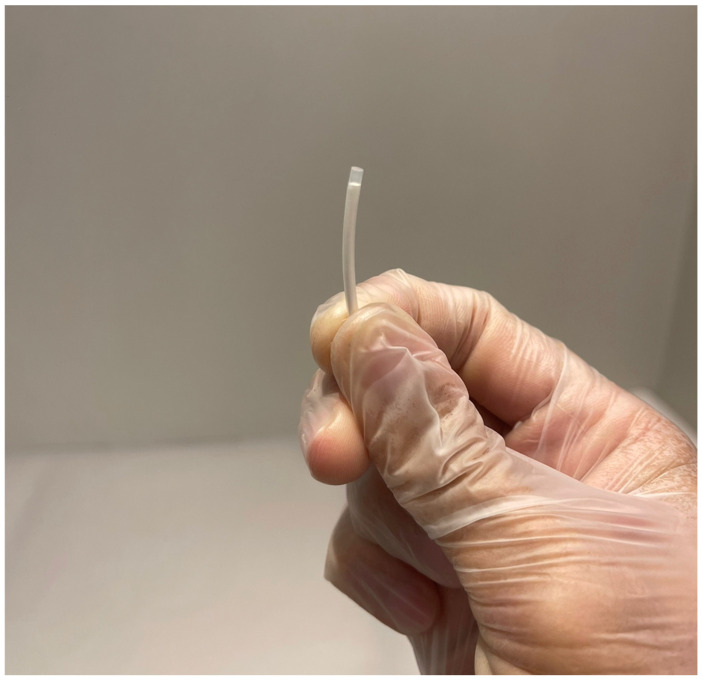
SA subdermal silastic implant.

**Figure 3 biomedicines-11-02586-f003:**
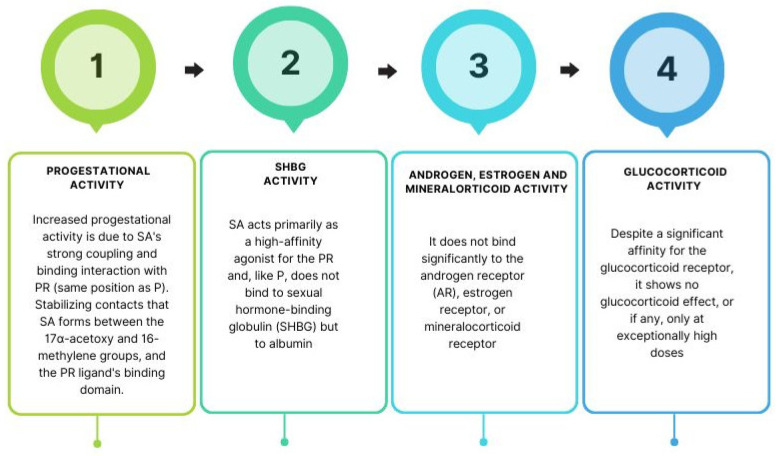
SA activity and mechanisms of action. (SA: segesterone acetate; PR: progesterone receptor; P: progesterone; SHBG: sexual hormone binding globulin; AR: androgen receptor).

**Table 1 biomedicines-11-02586-t001:** Safety, applications, and benefits of SA therapy in women.

Author(s), Year, Reference Number	SA Dose	Characteristics of Study/Participants	Results
Gemzell-Danielsson et al. (2019) [4]	CVS releasing an average of SA 150 mcg and EE 13 mcg daily.	A 12-month prospective study, including 3052 pre-menopausal women.	The 1-year SA/EE CVS has an acceptable safety profile.
Weisberg et al. (2005) [5]	CVS releasing 50 mcg/day of SA and either 10 (50/10) or 20 (50/20) mcg/day of EE.	A 6-month prospective study, two-stage trial, including 246 pre-menopause women.	CVS used with a bleeding-signaled regimen led to few terminations attributed to bleeding problems and to acceptable continuation rates.
Brache et al. (2015) [11]	Transdermal gel with doses of 1.5 mg SA/0.5 mg E2, 3.0 mg SA/1.0 mg E2, and 4.5 mg SA/1.5 mg E2). Participants applied gel daily on the abdomen for 21 consecutive days.	A randomized, open-label, three-treatment-period cross-over study, including 18 pre-menopausal women, to evaluate the effects of SA/E2 transdermal gel.	All three doses of SA/E2 transdermal gel blocked ovulation effectively with good safety and acceptable profile.
Lähteenmäki et al. (1985) [13]	Contraceptive SA subcutaneous silastic implant containing 20 mg or 40 mg.	A prospective study with 18 pre-menopausal women.	Both single subcutaneous implants of SA with 20 or 40 mg cause ovulation inhibition, decreasing the plasma LH/FSH ratio.
Coutinho et al. (1981) [17]	Contraceptive SA subcutaneous silastic implant containing 35 mg.	A prospective study with 282 pre-menopausal women.	The single subcutaneous implant of SA presents full protection for a period of 6 months or longer, providing an alternative for long-acting injectable contraceptives.
Archer et al. (2019) [21]	CVS designed to last 1 year (13 cycles), delivering a daily average of 0.15 mg SA and 0.013 mg EE.	A multicentre, open-label, single-arm study with 2278 pre-menopausal women.	The SA and EE CVS is an effective contraceptive for 13 consecutive cycles of use with an excellent safety profile.
Coutinho et al. (1999) [23]	Contraceptive SA subcutaneous silastic implant containing 50 mg.	A prospective study with 135 pre-menopausal women (66 breast-feeding women receiving SA implants and 69 women who chose to use Copper-T380 IUD as control subjects).	A single SA silastic implant placed subcutaneously at 6-month intervals appears to be an effective method of contraception for lactating women and results in blood concentrations of nursing infants at or near undetectable levels.
Silvin et al. (2005) [25]	Contraceptive SA subcutaneous silastic implant containing 92.7 mg ± 2.4 mg.	A 2-year multicentre prospective study enrolling 300 pre-menopausal women in Latin America.	The 2-year cumulative pregnancy rate was 1.7 per 100, with a Pearl index of 0.6 per 100 for the 2-year period. The 1-year and 2-year continuation rates were 80.5 and 66.7 per 100, respectively.
Ylänen et al. (2003) [34]	Two to four SA implants were inserted subcutaneously for 7 months. Each implant contained 15 mg, 20 mg, or 40 mg of SA.	A prospective study with 21 pre-menopausal women in whom endometriosis was diagnosed and treated.	All three doses of SA silastic implant effectively alleviated endometriosis-related pain. No significant changes in the serum levels of total, HDL, or LDL cholesterol were observed.

CVS: contraceptive vaginal system; EE: ethinyl estradiol; E2: estradiol; LH: luteinizing hormone; FSH: follicle-stimulating hormone; IUD: intrauterine devices; HDL: high-density lipoprotein; LDL: low-density lipoprotein.

**Table 2 biomedicines-11-02586-t002:** Possible clinical applications of SA.

Application	Ref.
Neuroprotection	[43,44,45,46,47]
Female Contraception	[1,2,3,4,5,6,7,12,21,22,23,24,25,26,48,49,50]
Male Contraception	[27,28,29,30,31,32,33]
PCOS	[3,21,26]
Menopause	[19,20,35,36,37,38,39,40,41,42,43,44]
Endometriosis	[1,3,34]

SA: segesterone acetate; PCOS: Polycystic Ovaries Syndrome.

## Data Availability

The datasets used and/or analyzed during the current study are available from the corresponding author upon reasonable request.

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
