# Peer review of "Clinical Approaches to Nestorone Subdermal Implant Therapy in Women’s Health"

_biomedicines, 2023, doi:10.3390/biomedicines11092586_

Round 1

Reviewer 1 Report

The manuscript presented by the authors raises a very interesting topic. The manuscript is clearly written and covers the most important issues of current gynecology. However, I have a few small reservations:

1. I believe that adding a "discussion" section to the manuscript in which the authors would present their own thoughts on this drug and possible new research directions. I think it would significantly increase the value of the work. However, I leave it to the authors' decision.

2. I suggest that the authors consider adding to the manuscript a diagram showing the possible mechanism of action of the drug. This could significantly improve the quality of the publication.

Author Response

REVIEWER 1

The manuscript presented by the authors raises a very interesting topic. The manuscript is clearly written and covers the most important issues of current gynecology. However, I have a few small reservations:

  1. “I believe that adding a "discussion" section to the manuscript in which the authors would present their own thoughts on this drug and possible new research directions. I think it would significantly increase the value of the work. However, I leave it to the authors' decision.”

A1: We thank the reviewer for drawing attention to this point and for the opportunity to clarify it. We agreed with the reviewer, and a discussion section was added to the manuscript.

  1. I suggest that the authors consider adding to the manuscript a diagram showing the possible mechanism of action of the drug. This could significantly improve the quality of the publication.

A2: We thank the reviewer for drawing attention to this point and for the opportunity to clarify it. We agree with the reviewer, and the drug mechanism of action was added, as shown in Figure 3.

Reviewer 2 Report

Comments about the manuscript:

“New Approaches to Nestorone Implant Therapy: from Contraception to Menopause as a Neuroprotective Drug”

Segesterone acetate (SA) or Nestorone is a synthetic progestin substance, devoid of androgenic, glucocorticoid or anabolic effects, which is administered in particular by the subcutaneous route. This type of implant can also be used for female contraception, post-menopausal hormonal treatment. Works have also shown that it could also be used in other cases (endometriosis, polycystic ovary syndrome) and that it could exhibit neuroprotective activity. However, the long-term safety of SA implant treatment remains controversial. The review presented here concerns the possible indications, doses, limitations and side effects of SA implant therapy.

Taking into account the medical importance of the subject treated, this interesting review is useful. This manuscript could be published after some very minor improvements.

Page 1, line 42, table 1. Include table 1 in the main text and not in the supplemental material.

References : Check the references, which is often a source of errors.

Author Response

REVIEWER 2

Segesterone acetate (SA) or Nestorone is a synthetic progestin substance, devoid of androgenic, glucocorticoid or anabolic effects, which is administered in particular by the subcutaneous route. This type of implant can also be used for female contraception, post-menopausal hormonal treatment. Works have also shown that it could also be used in other cases (endometriosis, polycystic ovary syndrome) and that it could exhibit neuroprotective activity. However, the long-term safety of SA implant treatment remains controversial. The review presented here concerns the possible indications, doses, limitations and side effects of SA implant therapy. Taking into account the medical importance of the subject treated, this interesting review is useful. This manuscript could be published after some very minor improvements.

  1. Page 1, line 42, table 1. Include table 1 in the main text and not in the supplemental material.

A3: We thank the reviewer for drawing attention to this point and for the opportunity to clarify it. We agree with the reviewer; Table 1 (actual Table 2) was included in the main text in Section 2. Discussion.

  1. References : Check the references, which is often a source of errors.

A4: We thank the reviewer for drawing attention to this point and for the opportunity to clarify it. We agree with the reviewer, and the references were checked and updated.

Reviewer 3 Report

In this manuscript, the authors analyze the possible indications, doses, limitations, and side effects of SA implant therapy. The review is well written and clear, but the authors should highlight, in a table, for each discussed application, the lower and higher effective used and tolerated concentrations, and their side effects.

They never mention the radius of action and possible side effects on the thyroid or other organs nearby the application point.

Do the effectiveness and the side effects vary with the patient's age?

Minor editing of the English language is required, and some typos to check.

Author Response

REVIEWER 3

  1. In this manuscript, the authors analyze the possible indications, doses, limitations, and side effects of SA implant therapy. The review is well written and clear, but the authors should highlight, in a table, for each discussed application, the lower and higher effective used and tolerated concentrations, and their side effects.

A5: We thank the reviewer for drawing attention to this point and for the opportunity to clarify it. We agree with the review and added table 1 with application doses in the limitations and adverse effects section.

  1. They never mention the radius of action and possible side effects on the thyroid or other organs nearby the application point.

A6: We thank the reviewer for drawing attention to this point and for the opportunity to clarify it. So far, studies of the impact of Nestorone on the metabolism of the thyroid gland or other organs have yet to be described. The implant does not cause any changes to the organs close to the application site due to its intradermal position and slow hormone release.

  1. Do the effectiveness and the side effects vary with the patient's age?

A7: We thank the reviewer for drawing attention to this point and for the opportunity to clarify it. Effectiveness doesn’t vary, but side effects are different. We added a paragraph on line 318 concerning this observation.

Reviewer 4 Report

Thank you for your submission.

The title is misleading and inaccurate, and can be misinterpreted. It implies that the drug has a role in menopause as a neuroprotective agent, but there is no clinical evidence for this.

There has been a published brief review of the Annovera contraceptive decice (Lee AL. Segesterone Acetate and Ethinyl Estradiol Vaginal Ring (Annovera) for Contraception. Am Fam Physician 2020;101:618-20). The authors concluded that “Annovera is an effective combined progestin/estrogen contraceptive device that can be used for one year. It could be considered a convenient option for women who desire a regular menstrual period without requiring medication refills from the pharmacy, but who do not want to use long-acting reversible contraception. However, the safety data on this new progesterone are limited, and it is less effective and more expensive than long-acting reversible contraception methods.

Other relevant reviews that are not mentioned are:

·      Nelson AL. Comprehensive overview of the recently FDA-approved contraceptive vaginal ring releasing segesterone acetate and ethinylestradiol: A new year-long, patient controlled, reversible birth control method. Expert Rev Clin Pharmacol 2019;12:953-63.

·      Paton DM. Contraceptive vaginal ring containing segesterone acetate and ethinyl estradiol: long-acting, patient-controlled, procedure-free, reversible prescription birth control. Drugs Today (Barc) 2019;55:449-57.

The references and information relating to the possible neuroprotective effects of progestogens are old and should be updated (e.g. Castelnovo LF, Thomas P. Progesterone exerts a neuroprotective action in a Parkinson's disease human cell model through membrane progesterone receptor alpha (mPRalpha/PAQR7). Front Endocrinol 2023;14:1125962).

The authors also need to justify whether SA would be expected to be any more effective than other progestogens (i.e. looks to be a drug class effect).

The English is mainly acceptable.

Author Response

REVIEWER 4

  1. The title is misleading and inaccurate, and can be misinterpreted. It implies that the drug has a role in menopause as a neuroprotective agent, but there is no clinical evidence for this.

A8. We thank the reviewer for drawing attention to this point and for the opportunity to clarify it. We disagree with the reviewer. The title encourages research on the use of Nestorone in menopause, where there are already described studies and our clinical experience. Changing the title may compromise the scientific impact of the manuscript and reduce the number of citations.

  1. There has been a published brief review of the Annovera contraceptive decice (Lee AL. Segesterone Acetate and Ethinyl Estradiol Vaginal Ring (Annovera) for Contraception. Am Fam Physician 2020;101:618-20). The authors concluded that “Annovera is an effective combined progestin/estrogen contraceptive device that can be used for one year. It could be considered a convenient option for women who desire a regular menstrual period without requiring medication refills from the pharmacy, but who do not want to use long-acting reversible contraception. However, the safety data on this new progesterone are limited, and it is less effective and more expensive than long-acting reversible contraception methods.”

A9. We thank the reviewer for drawing attention to this point and for the opportunity to clarify it. We agree with the reviewer, and this review was added to the manuscript and references list.

  1. Other relevant reviews that are not mentioned are:

  • Nelson AL. Comprehensive overview of the recently FDA-approved contraceptive vaginal ring releasing segesterone acetate and ethinylestradiol: A new year-long, patient controlled, reversible birth control method. Expert Rev Clin Pharmacol 2019;12:953-63.

  • Paton DM. Contraceptive vaginal ring containing segesterone acetate and ethinyl estradiol: long-acting, patient-controlled, procedure-free, reversible prescription birth control. Drugs Today (Barc) 2019;55:449-57.

A10. We thank the reviewer for drawing attention to this point and for the opportunity to clarify it. We agree with the reviewer, and these reviews were added to the manuscript and references list.

  1. The references and information relating to the possible neuroprotective effects of progestogens are old and should be updated (e.g. Castelnovo LF, Thomas P. Progesterone exerts a neuroprotective action in a Parkinson's disease human cell model through membrane progesterone receptor alpha (mPRalpha/PAQR7). Front Endocrinol 2023;14:1125962).

A11. We thank the reviewer for drawing attention to this point and for the opportunity to clarify it. We agree with the reviewer, and this study was added to the manuscript and references list.

  1. The authors also need to justify whether SA would be expected to be any more effective than other progestogens (i.e. looks to be a drug class effect).

A11. We thank the reviewer for drawing attention to this point and for the opportunity to clarify it. We agree with the reviewer, and regarding this, we added to the manuscript two figures: Figure 1 (Characteristics of synthetic progestin SA) and Figure 3 (SA activity and mechanisms of action).

Round 2

Reviewer 3 Report

The Authors accurately replied to all my concerns. In this status, the manuscript can be published in this journal.

Author Response

  1. The Authors accurately replied to all my concerns. In this status, the manuscript can be published in this journal.

A1. We thank the reviewer, and we appreciate the feedback. 

Reviewer 4 Report

As before, the title is misleading and inaccurate, and can be misinterpreted. It implies that the drug has a role in menopause as a neuroprotective agent, but there is no clinical evidence for this.

The authors' response is poor, indicating their interest in sensationalism and citations above scientific accuracy.

The work is very similar to Tuazon, JP; Sitruk-Ware, Regine. Beyond contraception and hormone replacement therapy: Advancing Nestorone to a 468 neuroprotective drug in the clinic. Brain Research 2019, 1704(), 161–163.

Nothing is presented to suggest that SA would be expected to be any more effective than other progestogens (i.e. looks to be a drug class effect).

Largely fine.

Author Response

  1. As before, the title is misleading and inaccurate, and can be misinterpreted. It implies that the drug has a role in menopause as a neuroprotective agent, but there is no clinical evidence for this.

A1: We thank the reviewer for drawing attention to this point and for the opportunity to clarify it. To comply with the reviewer's request, we changed the title of the manuscript to Clinical Approaches to Nestorone Subdermal Implant Therapy in Women's Health

  1. The authors' response is poor, indicating their interest in sensationalism and citations above scientific accuracy.

A2. We thank the reviewer for drawing attention to this point and for the opportunity to clarify it. We respond to all reviewer requests. We are not interested in sensationalism or propagating unscientific information.

  1. The work is very similar to Tuazon, JP; Sitruk-Ware, Regine. Beyond contraception and hormone replacement therapy: Advancing Nestorone to a 468 neuroprotective drug in the clinic. Brain Research 2019, 1704(), 161–163.

A3. We thank the reviewer for drawing attention to this point and for the opportunity to clarify it. Despite the similarity, the article by Tuazon et al is extremely concise and does not comprehensively address the possible uses of Nestorone. Furthermore, the article referred to is very small, only 3 pages long, and only addresses the issue of the possible neuroprotective effect of Nestorone. Our article is complete and covers most of the effects of Nestorone.

  1. Nothing is presented to suggest that SA would be expected to be any more effective than other progestogens (i.e. looks to be a drug class effect).

A4. We thank the reviewer for drawing attention to this point and for the opportunity to clarify it. Our review aims to evaluate the effects of nestorone. At no point in the manuscript did we make a favorable comparison to nestorone about other progestins. We brought what exists in the literature and future perspectives about Nestorone.

Round 3

Reviewer 4 Report

Thank you for the revision

It is fine, thanks.